# Inference in Graphical Models
# via Semidefinite Programming Hierarchies

**Murat A. Erdogdu**
Microsoft Research
erdogdu@cs.toronto.edu

**Yash Deshpande**
MIT and Microsoft Research
yash@mit.edu

**Andrea Montanari**
Stanford University
montanari@stanford.edu

## Abstract

Maximum A posteriori Probability (MAP) inference in graphical models amounts to solving a graph-structured combinatorial optimization problem. Popular inference algorithms such as belief propagation (BP) and generalized belief propagation (GBP) are intimately related to linear programming (LP) relaxation within the Sherali-Adams hierarchy. Despite the popularity of these algorithms, it is well understood that the Sum-of-Squares (SOS) hierarchy based on semidefinite programming (SDP) can provide superior guarantees. Unfortunately, SOS relaxations for a graph with $n$ vertices require solving an SDP with $n^{\Theta(d)}$ variables where $d$ is the degree in the hierarchy. In practice, for $d \geq 4$, this approach does not scale beyond a few tens of variables. In this paper, we propose binary SDP relaxations for MAP inference using the SOS hierarchy with two innovations focused on computational efficiency. Firstly, in analogy to BP and its variants, we only introduce decision variables corresponding to contiguous regions in the graphical model. Secondly, we solve the resulting SDP using a non-convex Burer-Monteiro style method, and develop a sequential rounding procedure. We demonstrate that the resulting algorithm can solve problems with tens of thousands of variables within minutes, and outperforms BP and GBP on practical problems such as image denoising and Ising spin glasses. Finally, for specific graph types, we establish a sufficient condition for the tightness of the proposed partial SOS relaxation.

## 1   Introduction

Graphical models provide a powerful framework for analyzing systems comprised by a large number of interacting variables. Inference in graphical models is crucial in scientific methodology with countless applications in a variety of fields including causal inference, computer vision, statistical physics, information theory, and genome research [WJ08, KF09, MM09].

In this paper, we propose a class of inference algorithms for pairwise undirected graphical models. Such models are fully specified by assigning: $(i)$ a finite domain $\mathcal{X}$ for the variables; $(ii)$ a finite graph $G = (V, E)$ for $V = [n] \equiv \{1, \dots, n\}$ capturing the interactions of the basic variables; $(iii)$ a collection of functions $\boldsymbol{\theta} = (\{\theta_i^v\}_{i \in V}, \{\theta_{ij}^e\}_{(i,j) \in E})$ that quantify the vertex potentials and interactions between the variables; whereby for each vertex $i \in V$ we have $\theta_i^v : \mathcal{X} \to \mathbb{R}$ and for each edge $(i, j) \in E$, we have $\theta_{ij}^e : \mathcal{X} \times \mathcal{X} \to \mathbb{R}$ (an arbitrary ordering is fixed on the pair of vertices $\{i, j\}$). These parameters can be used to form a probability distribution on $\mathcal{X}^V$ for the random vector $\boldsymbol{x} = (x_1, x_2, ..., x_n) \in \mathcal{X}^V$ by letting,

$$p(\boldsymbol{x}|\boldsymbol{\theta}) = \frac{1}{Z(\boldsymbol{\theta})} \, e^{U(\boldsymbol{x};\boldsymbol{\theta})}, \quad U(\boldsymbol{x};\boldsymbol{\theta}) = \sum_{(i,j)\in E} \theta_{ij}^e(x_i, x_j) + \sum_{i \in V} \theta_i^v(x_i), \quad (1.1)$$

where $Z(\boldsymbol{\theta})$ is the normalization constant commonly referred to as the partition function. While such models can encode a rich class of multivariate probability distributions, basic inference tasks are

intractable except for very special graph structures such as trees or small treewidth graphs [CD$^+$06]. In this paper, we will focus on MAP estimation, which amounts to solving the combinatorial optimization problem

$$\hat{\boldsymbol{x}}(\boldsymbol{\theta}) \equiv \arg \max_{\boldsymbol{x} \in \mathcal{X}^V} U(\boldsymbol{x}; \boldsymbol{\theta}). \tag{1.2}$$

Intractability plagues other classes of graphical models as well (e.g. Bayesian networks, factor graphs), and has motivated the development of a wide array of heuristics. One of the simplest such heuristics is the loopy belief propagation (BP) [WJ08, KF09, MM09]. In its max-product version (that is well-suited for MAP estimation), BP is intimately related to the linear programming (LP) relaxation of the combinatorial problem $\max_{\boldsymbol{x} \in \mathcal{X}^V} U(\boldsymbol{x}; \boldsymbol{\theta})$. Denoting the decision variables by $\boldsymbol{b} = (\{b_i\}_{i \in V}, \{b_{ij}\}_{(i,j) \in E})$, LP relaxation form of BP can be written as

$$\underset{\boldsymbol{b}}{\text{maximize}} \quad \sum_{(i,j) \in E} \sum_{x_i, x_j \in \mathcal{X}} \theta_{ij}(x_i, x_j) b_{ij}(x_i, x_j) + \sum_{i \in V} \sum_{x_i \in \mathcal{X}} \theta_i(x_i) b_i(x_i), \tag{1.3}$$

$$\text{subject to} \quad \sum_{x_j \in \mathcal{X}} b_{ij}(x_i, x_j) = b_i(x_i) \qquad \forall (i,j) \in E, \tag{1.4}$$

$$b_i \in \Delta_{\mathcal{X}} \quad \forall i \in V, \quad b_{ij} \in \Delta_{\mathcal{X} \times \mathcal{X}} \quad \forall (i,j) \in E, \tag{1.5}$$

where $\Delta_S$ denotes the simplex of probability distributions over set $S$. The decision variables are referred to as 'beliefs', and their feasible set is a relaxation of the polytope of marginals of distributions. The beliefs satisfy the constraints on marginals involving at most two variables connected by an edge.

Loopy belief propagation is successful on some applications, e.g. sparse locally tree-like graphs that arise, for instance, decoding modern error correcting codes [RU08] or in random constraint satisfaction problems [MM09]. However, in more structured instances – arising for example in computer vision – BP can be substantially improved by accounting for local dependencies within subsets of more than two variables. This is achieved by generalized belief propagation (GBP) [YFW05] where the decision variables are beliefs $b_R$ that are defined on subsets of vertices (a 'region') $R \subseteq [n]$, and that represent the marginal distributions of the variables in that region. The basic constraint on the beliefs is the linear marginalization constraint: $\sum_{\boldsymbol{x}_{R \setminus S}} b_R(\boldsymbol{x}_R) = b_S(\boldsymbol{x}_S)$, holding whenever $S \subseteq R$. Hence GBP itself is closely related to LP relaxation of the polytope of marginals of probability distributions. The relaxation becomes tighter as larger regions are incorporated. In a prototypical application, $G$ is a two-dimensional grid, and regions are squares induced by four contiguous vertices (plaquettes), see Figure 1, left frame. Alternatively in the right frame of the same figure, the regions correspond to triangles.

The LP relaxations that correspond to GBP are closely related to the Sherali-Adams hierarchy [SA90]. Similar to GBP, the variables within this hierarchy are beliefs over subsets of variables $\boldsymbol{b}_R = (b_R(\boldsymbol{x}_R))_{\boldsymbol{x}_R \in \mathcal{X}^R}$ which are consistent under marginalization: $\sum_{\boldsymbol{x}_{R \setminus S}} b_R(\boldsymbol{x}_R) = b_S(\boldsymbol{x}_S)$. However, these two approaches differ in an important point: Sherali-Adams hierarchy uses beliefs over *all subsets* of $|R| \le d$ variables, where $d$ is the degree in the hierarchy; this leads to an LP of size $\Theta(n^d)$. In contrast, GBP only retains regions that are contiguous in $G$. If $G$ has maximum degree $k$, this produces an LP of size $\mathcal{O}(nk^d)$, a reduction which is significant for large-scale problems.

Given the broad empirical success of GBP, it is natural to develop better methods for inference in graphical models using tighter convex relaxations. Within combinatorial optimization, it is well understood that the semidefinite programming (SDP) relaxations provide superior approximation guarantees with respect to LP [GW95]. Nevertheless, SDP has found limited applications in inference tasks for graphical models for at least two reasons. A *structural reason*: standard SDP relaxations (e.g. [GW95]) do not account exactly for correlations between neighboring vertices in the graph which is essential for structured graphical models. As a consequence, BP or GBP often outperforms basic SDPs. A *computational reason*: basic SDP relaxations involve $\Theta(n^2)$ decision variables, and generic interior point solvers do not scale well for the large-scale applications. An exception is [WJ04] which employs the simplest SDP relaxation (degree 2 Sum-Of-Squares, see below) in conjunction with a relaxation of the entropy and interior point methods – higher order relaxations are briefly discussed without implementation as the resulting program suffers from the aforementioned limitations.

In this paper, we revisit MAP inference in graphical models via SDPs, and propose an approach that carries over the favorable performance guarantees of SDPs into inference tasks. For simplicity, we focus on models with binary variables, but we believe that many of the ideas developed here can be naturally extended to other finite domains. We present the following contributions:

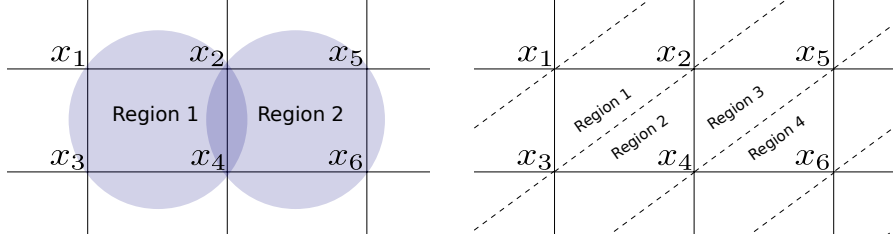

Figure 1: A two dimensional grid, and two typical choices for regions for GBP and PSOS. Left: Regions are plaquettes comprising four vertices. Right: Regions are triangles.

**Partial Sum-Of-Squares relaxations.** We use SDP hierarchies, specifically the Sum-Of-Squares (SOS) hierarchy [Sho87, Las01, Par03] to formulate tighter SDP relaxations for binary MAP inference that account exactly for the joint distributions of small subsets of variables $\boldsymbol{x}_R$, for $R \subseteq V$. However, SOS introduces decision variables for all subsets $R \subseteq V$ with $|R| \leq d/2$ ($d$ is a fixed even integer), and hence scales poorly for large-scale inference problems. We propose a similar modification as in GBP. Instead of accounting for all subsets $R$ with $|R| \leq d/2$, we only introduce decision variables to represent a certain family of such subsets (regions) of vertices in $G$. The resulting SDP has (for $d$ and the maximum degree of $G$ bounded) only $\mathcal{O}(n^2)$ decision variables which is suitable for practical implementations. We refer to these relaxations as Partial Sum-Of-Squares (PSOS), cf. Section 2.

**Theoretical analysis.** In Section 2.1, we prove that suitable PSOS relaxations are tight for certain classes of graphs, including planar graphs, with $\theta^{\mathrm{v}} = 0$. While this falls short of explaining the empirical results (which uses simpler relaxations, and $\theta^{\mathrm{v}} \neq 0$), it points in the right direction.

**Optimization algorithm and rounding.** Despite the simplification afforded by PSOS, interior-point solvers still scale poorly to large instances. In order to overcome this problem, we adopt a non-convex approach proposed by Burer and Monteiro [BM03]. We constrain the rank of the SDP matrix in PSOS to be at most $r$, and solve the resulting non-convex problem using a trust-region coordinate ascent method, cf. Section 3.1. Further, we develop a rounding procedure called Confidence Lift and Project (CLAP) which iteratively uses PSOS relaxations to obtain an integer solution, cf. Section 3.2.

**Numerical experiments.** In Section 4, we present numerical experiments with PSOS by solving problems of size up to $10,000$ within several minutes. While additional work is required to scale this approach to massive sizes, we view this as an exciting proof-of-concept. To the best of our knowledge, no earlier attempt was successful in scaling higher order SOS relaxations beyond tens of dimensions. More specifically, we carry out experiments with two-dimensional grids – an image denoising problem, and Ising spin glasses. We demonstrate through extensive numerical studies that PSOS significantly outperforms BP and GBP in the inference tasks we consider.

## 2 Partial Sum-Of-Squares Relaxations

For concreteness, throughout the paper we focus on pairwise models with binary variables. We do not expect fundamental problems extending the same approach to other domains. For binary variables $\boldsymbol{x} = (x_1, x_2, ..., x_n)$, MAP estimation amounts to solving the following optimization problem

$$\underset{\boldsymbol{x}}{\text{maximize}} \quad \sum_{(i,j) \in E} \theta_{ij}^{\mathrm{e}} x_i x_j + \sum_{i \in V} \theta_i^{\mathrm{v}} x_i \,, \tag{INT}$$
$$\text{subject to} \quad x_i \in \{+1, -1\} \,, \quad \forall i \in V \,,$$

where $\theta^{\mathrm{e}} = (\theta_{ij}^{\mathrm{e}})_{1 \leq i,j \leq n}$ and $\theta^{\mathrm{v}} = (\theta_i^{\mathrm{v}})_{1 \leq i \leq n}$ are the parameters of the graphical model.

For the reader's convenience, we recall a few basic facts about SOS relaxations, referring to [BS16] for further details. For an even integer $d$, $\mathrm{SOS}(d)$ is an SDP relaxation of INT with decision variable $X : \binom{[n]}{\leq d} \to \mathbb{R}$ where $\binom{[n]}{\leq d}$ denotes the set of subsets $S \subseteq [n]$ of size $|S| \leq d$; it is given as

$$\underset{X}{\text{maximize}} \quad \sum_{(i,j) \in E} \theta_{ij}^{\mathrm{e}} X(\{i,j\}) + \sum_{i \in V} \theta_i^{\mathrm{v}} X(\{i\}) \,, \tag{SOS}$$
$$\text{subject to} \quad X(\emptyset) = 1, \quad \mathsf{M}(X) \succcurlyeq 0 \,.$$

The moment matrix $\mathsf{M}(X)$ is indexed by sets $S, T \subseteq [n]$, $|S|, |T| \leq d/2$, and has entries $\mathsf{M}(X)_{S,T} = X(S \triangle T)$ with $\triangle$ denoting the symmetric difference of two sets. Note that $\mathsf{M}(X)_{S,S} = X(\emptyset) = 1$.

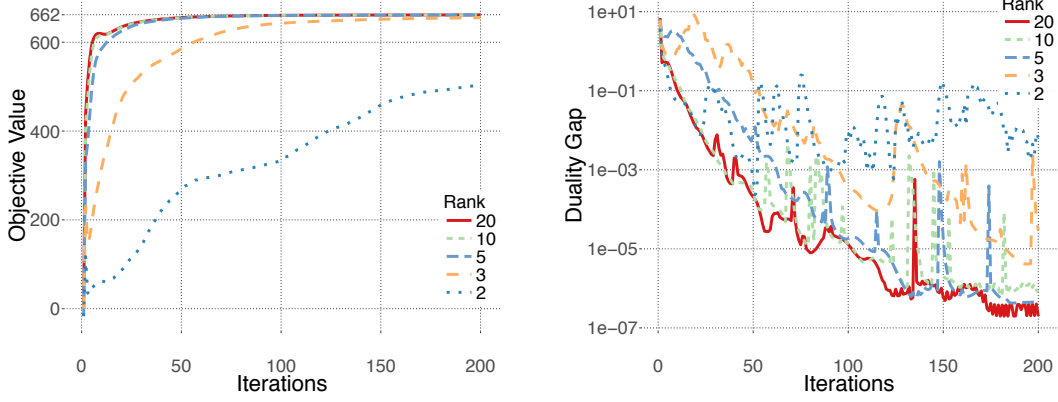

Figure 2: Effect of the rank constraint $r$ on $n = 400$ square lattice ($20 \times 20$): Left plot shows the change in the value of objective at each iteration. Right plot shows the duality gap of the Lagrangian.

We can equivalently represent $\mathsf{M}(X)$ as a Gram matrix by letting $\mathsf{M}(X)_{S,T} = \langle \boldsymbol{\sigma}_S, \boldsymbol{\sigma}_T \rangle$ for a collection of vectors $\boldsymbol{\sigma}_S \in \mathbb{R}^r$ indexed by $S \in \binom{[n]}{\leq d/2}$. The case $r = |\binom{[n]}{\leq d/2}|$ can represent any semidefinite matrix; however, in what follows it is convenient from a computational perspective to consider smaller choices of $r$. The constraint $\mathsf{M}(X)_{S,S} = 1$ is equivalent to $\|\boldsymbol{\sigma}_S\| = 1$, and the condition $M(X)_{S,T} = X(S \triangle T)$ can be equivalently written as

$$\langle \boldsymbol{\sigma}_{S_1}, \boldsymbol{\sigma}_{T_1} \rangle = \langle \boldsymbol{\sigma}_{S_2}, \boldsymbol{\sigma}_{T_2} \rangle, \qquad \forall S_1 \triangle T_1 = S_2 \triangle T_2. \tag{2.1}$$

In the case $d = 2$, SOS(2) recovers the classical Goemans-Williamson SDP relaxation [GW95].

In the following, we consider the simplest higher-order SDP, namely SOS(4) for which the general constraints in Eq. (2.1) can be listed explicitly. Fixing a region $R \subseteq V$, and defining the Gram vectors $\boldsymbol{\sigma}_\emptyset, (\boldsymbol{\sigma}_i)_{i \in V}, (\boldsymbol{\sigma}_{ij})_{\{i,j\} \subseteq V}$, we list the constraints that involve vectors $\boldsymbol{\sigma}_S$ for $S \subseteq R$ and $|S| = 1, 2$:

$$\begin{aligned}
\|\boldsymbol{\sigma}_i\| &= 1 & \forall i \in S \cup \{\emptyset\}, & \qquad \text{(Sphere } \textcircled{i}\text{)} \\
\langle \boldsymbol{\sigma}_i, \boldsymbol{\sigma}_j \rangle &= \langle \boldsymbol{\sigma}_{ij}, \boldsymbol{\sigma}_\emptyset \rangle & \forall i, j \in S, & \qquad \text{(Undirected } i - j\text{)} \\
\langle \boldsymbol{\sigma}_i, \boldsymbol{\sigma}_{ij} \rangle &= \langle \boldsymbol{\sigma}_j, \boldsymbol{\sigma}_\emptyset \rangle & \forall i, j \in S, & \qquad \text{(Directed } i \to j\text{)} \\
\langle \boldsymbol{\sigma}_i, \boldsymbol{\sigma}_{jk} \rangle &= \langle \boldsymbol{\sigma}_k, \boldsymbol{\sigma}_{ij} \rangle & \forall i, j, k \in S, & \qquad \text{(V-shaped } {}^i_j V^k\text{)} \\
\langle \boldsymbol{\sigma}_{ij}, \boldsymbol{\sigma}_{jk} \rangle &= \langle \boldsymbol{\sigma}_{ik}, \boldsymbol{\sigma}_\emptyset \rangle & \forall i, j, k \in S, & \qquad \text{(Triangle } {}_j\triangle_k^i\text{)} \\
\langle \boldsymbol{\sigma}_{ij}, \boldsymbol{\sigma}_{kl} \rangle &= \langle \boldsymbol{\sigma}_{ik}, \boldsymbol{\sigma}_{jl} \rangle & \forall i, j, k, l \in S. & \qquad \text{(Loop } {}^i_k\square^j_l\text{)}
\end{aligned}$$

Given an assignment of the Gram vectors $\boldsymbol{\sigma} = (\boldsymbol{\sigma}_\emptyset, (\boldsymbol{\sigma}_i)_{i \in V}, (\boldsymbol{\sigma}_{ij})_{\{i,j\} \subseteq V})$, we denote by $\boldsymbol{\sigma}|_R$ its restriction to $R$, namely $\boldsymbol{\sigma}|_R = (\boldsymbol{\sigma}_\emptyset, (\boldsymbol{\sigma}_i)_{i \in R}, (\boldsymbol{\sigma}_{ij})_{\{i,j\} \subseteq R})$. We denote by $\boldsymbol{\Omega}(R)$, the set of vectors $\boldsymbol{\sigma}|_R$ that satisfy the above constraints. With these notations, the SOS(4) SDP can be written as

$$\underset{\boldsymbol{\sigma}}{\text{maximize}} \quad \sum_{(i,j) \in E} \theta_{ij}^{\text{e}} \langle \boldsymbol{\sigma}_i, \boldsymbol{\sigma}_j \rangle + \sum_{i \in V} \theta_i^{\text{v}} \langle \boldsymbol{\sigma}_i, \boldsymbol{\sigma}_\emptyset \rangle, \tag{SOS(4)}$$

$$\text{subject to} \quad \boldsymbol{\sigma} \in \boldsymbol{\Omega}(V).$$

A specific Partial SOS (PSOS) relaxation is defined by a collection of regions $\mathcal{R} = \{R_1, R_2, \ldots, R_m\}$, $R_i \subseteq V$. We will require $\mathcal{R}$ to be a covering, i.e. $\cup_{i=1}^m R_i = V$ and for each $(i,j) \in E$ there exists $\ell \in [m]$ such that $\{i,j\} \subseteq R_\ell$. Given such a covering, the PSOS(4) relaxation is

$$\underset{\boldsymbol{\sigma}}{\text{maximize}} \quad \sum_{(i,j) \in E} \theta_{ij}^{\text{e}} \langle \boldsymbol{\sigma}_i, \boldsymbol{\sigma}_j \rangle + \sum_{i \in V} \theta_i^{\text{v}} \langle \boldsymbol{\sigma}_i, \boldsymbol{\sigma}_\emptyset \rangle, \tag{PSOS(4)}$$

$$\text{subject to} \quad \boldsymbol{\sigma}|_{R_i} \in \boldsymbol{\Omega}(R_i) \quad \forall i \in \{1, 2, \ldots, m\}.$$

Notice that variables $\boldsymbol{\sigma}_{ij}$ only enter the above program if $\{i, j\} \subseteq R_\ell$ for some $\ell$. As a consequence, the dimension of the above optimization problem is $\mathcal{O}(r \sum_{\ell=1}^m |R_\ell|^2)$, which is $\mathcal{O}(nr)$ if the regions have bounded size; this will be the case in our implementation. Of course, the specific choice of regions $\mathcal{R}$ is crucial for the quality of this relaxation. A natural heuristic is to choose each region $R_\ell$ to be a subset of contiguous vertices in $G$, which is generally the case for GBP algorithms.

---

**Algorithm 1:** `Partial-SOS`

---

**Input :** $G = (V, E)$, $\theta^e \in \mathbb{R}^{n \times n}$, $\theta^v \in \mathbb{R}^n$, $\boldsymbol{\sigma} \in \mathbb{R}^{r \times (1 + |V| + |E|)}$, Reliables $= \emptyset$

Actives $= V \cup E \setminus$ Reliables, and $\Delta = 1$,

**while** $\Delta > tol$ **do**

    $\Delta = 0$

    **for** $s \in$ Actives **do**

        **if** $s \in V$ **then**                                     `/* s ∈ V is a vertex */`

            $\boldsymbol{c}_s = \sum_{t \in \partial s} \theta^e_{st} \boldsymbol{\sigma}_t + \theta^v_s \boldsymbol{\sigma}_\emptyset$

        **else**                                         `/* s = (s₁, s₂) ∈ E is an edge */`

            $\boldsymbol{c}_s = \theta^e_{s_1 s_2} \boldsymbol{\sigma}_\emptyset + \theta^v_{s_1} \boldsymbol{\sigma}_{s_2} + \theta^v_{s_2} \boldsymbol{\sigma}_{s_1}$

        Form matrix $\boldsymbol{A}_s$, vector $\boldsymbol{b}_s$, and the corresponding Lagrange multipliers $\boldsymbol{\lambda}_s$ (see text).

        $\boldsymbol{\sigma}_s^{\text{new}} \longleftarrow \underset{\|\boldsymbol{\sigma}\|=1}{\arg\max} \left\{ \langle \boldsymbol{c}_s, \boldsymbol{\sigma} \rangle + \frac{\rho}{2} \|\boldsymbol{A}_s \boldsymbol{\sigma} - \boldsymbol{b}_s + \boldsymbol{\lambda}_s\|^2 \right\}$         `/* sub-problem */`

        $\Delta \longleftarrow \Delta + \|\boldsymbol{\sigma}_s^{\text{new}} - \boldsymbol{\sigma}_s\|^2 + \|\boldsymbol{A}_s \boldsymbol{\sigma}_s - \boldsymbol{b}_s\|^2$

        $\boldsymbol{\sigma}_s \longleftarrow \boldsymbol{\sigma}_s^{\text{new}}$                                         `/* update variables */`

        $\boldsymbol{\lambda}_s \longleftarrow \boldsymbol{\lambda}_s + \boldsymbol{A}_s \boldsymbol{\sigma}_s - \boldsymbol{b}_s$

---

## 2.1 Tightness guarantees

Solving exactly INT is NP-hard even if $G$ is a three-dimensional grid [Bar82]. Therefore, we do not expect PSOS(4) to be tight for general graphs $G$. On the other hand, in our experiments (cf. Section 4), PSOS(4) systematically achieves the exact maximum of INT for two-dimensional grids with random edge and vertex parameters $(\theta^e_{ij})_{(i,j) \in E}$, $(\theta^v_i)_{i \in V}$. This finding is quite surprising and calls for a theoretical explanation. While full understanding remains an open problem, we present here partial results in that direction.

Recall that a cycle in $G$ is a sequence of distinct vertices $(i_1, \ldots, i_\ell)$ such that, for each $j \in [\ell] \equiv \{1, 2, \ldots, \ell\}$, $(i_j, i_{j+1}) \in E$ (where $\ell + 1$ is identified with 1). The cycle is chordless if there is no $j, k \in [\ell]$, with $j - k \neq \pm 1 \mod \ell$ such that $(i_j, i_k) \in E$. We say that a collection of regions $\mathcal{R}$ on graph $G$ is *circular* if for each chordless cycle in $G$ there exists a region in $R \in \mathcal{R}$ such that all vertices of the cycle belong to $R$. We also need the following straightforward notion of contractibility. A *contraction* of $G$ is a new graph obtained by identifying two vertices connected by an edge in $G$. $G$ is *contractible* to $H$ if there exists a sequence of contractions transforming $G$ into $H$.

The following theorem is a direct consequence of a result of Barahona and Mahjoub [BM86] (see Supplement for a proof).

**Theorem 1.** *Consider the problem* INT *with* $\theta^v = 0$. *If $G$ is not contractible to $K_5$ (the complete graph over $5$ vertices), then* PSOS(4) *with a circular covering $\mathcal{R}$ is tight.*

The assumption that $\theta^v = 0$ can be made without loss of generality (see Supplement for the reduction from the general case). Furthermore, INT can be solved in polynomial time if $G$ is planar, and $\theta^v = 0$ [Bar82]. Note however, the reduction from $\theta^v \neq 0$ to $\theta^v = 0$ can transform a planar graph to a non-planar graph. This theorem implies that (full) SOS(4) is also tight if $G$ is not contractible to $K_5$. Notice that planar graphs are not contractible to $K_5$, and we recover the fact that INT can be solved in polynomial time if $\theta^v = 0$. This result falls short of explaining the empirical findings in Section 4, for at least two reasons. Firstly the reduction to $\theta^v = 0$ induces $K_5$ subhomomorphisms for grids. Second, the collection of regions $\mathcal{R}$ described in the previous section does not include all chordless cycles. Theoretically understanding the empirical performance of PSOS(4) as stated remains open. However, similar cycle constraints have proved useful in analyzing LP relaxations [WRS16].

## 3 Optimization Algorithm and Rounding

### 3.1 Solving PSOS(4) via Trust-Region Coordinate Ascent

We will approximately solve PSOS(4) while keeping $r = \mathcal{O}(1)$. Earlier work implies that (under suitable genericity condition on the SDP) there exists an optimal solution with rank $\sqrt{2 \, \# \, \text{constraints}}$ [Pat98]. Recent work [BVB16] shows that for $r > \sqrt{2 \, \# \, \text{constraints}}$, the non-convex optimization problem has no non-global local maxima. For SOS(2), [MM$^+$17] proves that setting $r = \mathcal{O}(1)$ is sufficient for achieving $\mathcal{O}(1/r)$ relative error from the global maximum for specific choices of potentials $\theta^e, \theta^v$. We find that there is little or no improvement beyond $r = 10$ (cf. Figure 2).

---
**Algorithm 2:** `CLAP: Confidence Lift And Project`

---
**Input :** $G = (V, E)$, $\theta^{\mathrm{e}} \in \mathbb{R}^{n \times n}$, $\theta^{\mathrm{v}} \in \mathbb{R}^n$, regions $\mathcal{R} = \{R_1, ..., R_m\}$

Initialize variable matrix $\boldsymbol{\sigma} \in \mathbb{R}^{r \times (1+|V|+|E|)}$ and set Reliables $= \emptyset$.

**while** Reliables $\neq V \cup E$ **do**

    Run `Partial-SOS` on inputs $G = (V, E)$, $\theta^{\mathrm{e}}$, $\theta^{\mathrm{v}}$, $\boldsymbol{\sigma}$, Reliables       /* lift procedure */

    Promotions $= \emptyset$ and Confidence $= 0.9$

    **while** Confidence $> 0$ **and** Promotions $\neq \emptyset$ **do**

        **for** $s \in V \cup E \setminus$ Reliables **do**             /* find promotions */

            **if** $|\langle \boldsymbol{\sigma}_\emptyset, \boldsymbol{\sigma}_s \rangle| >$ Confidence **then**

                $\boldsymbol{\sigma}_s = \mathrm{sign}(\langle \boldsymbol{\sigma}_\emptyset, \boldsymbol{\sigma}_s \rangle) \cdot \boldsymbol{\sigma}_\emptyset$        /* project procedure */

                Promotions $\longleftarrow$ Promotions $\cup \{s_c\}$

        **if** Promotions $= \emptyset$ **then**          /* decrease confidence level */

            Confidence $\longleftarrow$ Confidence $- 0.1$

        Reliables $\longleftarrow$ Reliables $\cup$ Promotions         /* update Reliables */

---
**Output :** $(\langle \boldsymbol{\sigma}_i, \boldsymbol{\sigma}_\emptyset \rangle)_{i \in V} \in \{-1, +1\}^n$

---

We will assume that $\mathcal{R} = (R_1, \ldots, R_m)$ is a covering of $G$ (in the sense introduced in the previous section), and –without loss of generality– we will assume that the edge set is

$$E = \left\{ (i, j) \in V \times V : \quad \exists \ell \in [m] \quad \text{such that} \quad \{i, j\} \subseteq R_\ell \right\}. \tag{3.1}$$

In other words, $E$ is the maximal set of edges that is compatible with $\mathcal{R}$ being a covering. This can always be achieved by adding new edges $(i, j)$ to the original edge set with $\theta_{ij}^{\mathrm{e}} = 0$. Hence, the decision variables $\boldsymbol{\sigma}_s$ are indexed by $s \in \mathcal{S} = \{\emptyset\} \cup V \cup E$. Apart from the norm constraints, all other consistency constraints take the form $\langle \boldsymbol{\sigma}_s, \boldsymbol{\sigma}_r \rangle = \langle \boldsymbol{\sigma}_t, \boldsymbol{\sigma}_p \rangle$ for some 4-tuple of indices $(s, r, t, p)$. We denote the set of all such 4-tuples by $\mathcal{C}$, and construct the augmented Lagrangian of PSOS(4) as

$$\mathcal{L}(\boldsymbol{\sigma}, \boldsymbol{\lambda}) = \sum_{i \in V} \theta_i^{\mathrm{v}} \langle \boldsymbol{\sigma}_i, \boldsymbol{\sigma}_\emptyset \rangle + \sum_{(i,j) \in E} \theta_{ij}^{\mathrm{e}} \langle \boldsymbol{\sigma}_i, \boldsymbol{\sigma}_j \rangle + \frac{\rho}{2} \sum_{(s,r,t,p) \in \mathcal{C}} \left( \langle \boldsymbol{\sigma}_s, \boldsymbol{\sigma}_r \rangle - \langle \boldsymbol{\sigma}_t, \boldsymbol{\sigma}_p \rangle + \lambda_{s,r,t,p} \right)^2.$$

At each step, our algorithm execute two operations: $(i)$ maximize the cost function with respect to one of the vectors $\boldsymbol{\sigma}_s$; $(ii)$ perform one step of gradient descent with respect to the corresponding subset of Lagrangian parameters, to be denoted by $\boldsymbol{\lambda}_s$. More precisely, fixing $s \in \mathcal{S} \setminus \{\emptyset\}$ (by rotational invariance, it is not necessary to update $\boldsymbol{\sigma}_\emptyset$), we note that $\boldsymbol{\sigma}_s$ appears in the constraints linearly (or it does not appear). Hence, we can write these constraints in the form $\boldsymbol{A}_s \boldsymbol{\sigma}_s = \boldsymbol{b}_s$ where $\boldsymbol{A}_s, \boldsymbol{b}_s$ depend on $(\boldsymbol{\sigma}_r)_{r \neq s}$ but not on $\boldsymbol{\sigma}_s$. We stack the corresponding Lagrangian parameters in a vector $\boldsymbol{\lambda}_s$; therefore the Lagrangian term involving $\boldsymbol{\sigma}_s$ reads $(\rho/2) \| \boldsymbol{A}_s \boldsymbol{\sigma}_s - \boldsymbol{b}_s + \boldsymbol{\lambda}_s \|^2$. On the other hand, the graphical model contribution is that the first two terms in $\mathcal{L}(\boldsymbol{\sigma}, \boldsymbol{\lambda})$ are linear in $\boldsymbol{\sigma}_s$, and hence they can be written as $\langle \boldsymbol{c}_s, \boldsymbol{\sigma}_s \rangle$. Summarizing, we have

$$\mathcal{L}(\boldsymbol{\sigma}, \boldsymbol{\lambda}) = \langle \boldsymbol{c}_s, \boldsymbol{\sigma}_s \rangle + \| \boldsymbol{A}_s \boldsymbol{\sigma}_s - \boldsymbol{b}_s + \boldsymbol{\lambda}_s \|^2 + \widetilde{\mathcal{L}}\big( (\boldsymbol{\sigma}_r)_{r \neq s}, \boldsymbol{\lambda} \big). \tag{3.2}$$

It is straightforward to compute $\boldsymbol{A}_s, \boldsymbol{b}_s, \boldsymbol{c}_s$; in particular, for $(s, r, t, p) \in \mathcal{C}$, the rows of $\boldsymbol{A}_s$ and $\boldsymbol{b}_s$ are indexed by $r$ such that the vectors $\boldsymbol{\sigma}_r$ form the rows of $\boldsymbol{A}_s$, and $\langle \boldsymbol{\sigma}_t, \boldsymbol{\sigma}_p \rangle$ form the corresponding entry of $\boldsymbol{b}_s$. Further, if $s$ is a vertex and $\partial s$ are its neighbors, we set $\boldsymbol{c}_s = \sum_{t \in \partial s} \theta_{st}^{\mathrm{e}} \boldsymbol{\sigma}_t + \theta_s^{\mathrm{v}} \boldsymbol{\sigma}_\emptyset$ while if $s = (s_1, s_2)$ is an edge, we set $\boldsymbol{c}_s = \theta_{s_1 s_2}^{\mathrm{e}} \boldsymbol{\sigma}_\emptyset + \theta_{s_1}^{\mathrm{v}} \boldsymbol{\sigma}_{s_2} + \theta_{s_2}^{\mathrm{v}} \boldsymbol{\sigma}_{s_1}$. Note that we are using the equivalent representations $\langle \boldsymbol{\sigma}_i, \boldsymbol{\sigma}_j \rangle = \langle \boldsymbol{\sigma}_{ij}, \boldsymbol{\sigma}_\emptyset \rangle$, $\langle \boldsymbol{\sigma}_{ij}, \boldsymbol{\sigma}_j \rangle = \langle \boldsymbol{\sigma}_i, \boldsymbol{\sigma}_\emptyset \rangle$, and $\langle \boldsymbol{\sigma}_{ij}, \boldsymbol{\sigma}_i \rangle = \langle \boldsymbol{\sigma}_j, \boldsymbol{\sigma}_\emptyset \rangle$.

Finally, we maximize Eq. (3.2) with respect to $\boldsymbol{\sigma}_s$ by a Moré-Sorenson style method [MS83].

### 3.2 Rounding via Confidence Lift and Project

After Algorithm 1 generates an approximate optimizer $\boldsymbol{\sigma}$ for PSOS(4), we reduce its rank to produce a solution of the original combinatorial optimization problem INT. To this end, we interpret $\langle \boldsymbol{\sigma}_i, \boldsymbol{\sigma}_\emptyset \rangle$ as our belief about the value of $x_i$ in the optimal solution of INT, and $\langle \boldsymbol{\sigma}_{ij}, \boldsymbol{\sigma}_\emptyset \rangle$ as our belief about the value of $x_i x_j$. This intuition can be formalized using the notion of pseudo-probability [BS16]. We then recursively round the variables about which we have strong beliefs; we fix rounded variables in the next iteration, and solve the induced PSOS(4) on the remaining ones.

More precisely, we set a confidence threshold Confidence. For any variable $\boldsymbol{\sigma}_s$ such that $|\langle \boldsymbol{\sigma}_s, \boldsymbol{\sigma}_\emptyset \rangle| >$ Confidence, we let $x_s = \mathrm{sign}(\langle \boldsymbol{\sigma}_s, \boldsymbol{\sigma}_\emptyset \rangle)$ and fix $\boldsymbol{\sigma}_s = x_s \boldsymbol{\sigma}_\emptyset$. These variables $\boldsymbol{\sigma}_s$ are no longer

| True | Noisy | BP-SP | BP-MP | GBP | PSOS(2) | PSOS(4) |
|------|-------|-------|-------|-----|---------|---------|
| 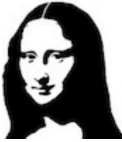 |  |  |  |  |  |  |

Bernoulli $p = 0.2$

| U(x): | 25815 | 19237 | 26165 | 26134 | 26161 | 26015 | 26194 |
| Time: | - | - | 2826s | 2150s | 7894s | 454s | 5059s |

|  |  |  |  |  |  |  |

Blockwise $p=0.006$

| U(x): | 27010 | 26808 | 27230 | 27012 | 27232 | 26942 | 27252 |
| Time: | - | - | 1674s | 729s | 8844s | 248s | 4457s |

Figure 3: Denoising a binary image by maximizing the objective function Eq. (4.1). Top row: i.i.d. Bernoulli error with flip probability $p = 0.2$ with $\theta_0 = 1.26$. Bottom row: blockwise noise where each pixel is the center of a $3 \times 3$ error block independently with probability $p = 0.006$ and $\theta_0 = 1$.

updated, and instead the reduced SDP is solved. If no variable satisfies the confidence condition, the threshold is reduced until variables are found that satisfy it. After the first iteration, most variables yield strong beliefs and are fixed; hence the consequent iterations have fewer variables and are faster.

## 4 Numerical Experiments

In this section, we validate the performance of the Partial SOS relaxation and the CLAP rounding scheme on models defined on two-dimensional grids. Grid-like graphical models are common in a variety of fields such as computer vision [SSZ02], and statistical physics [MM09]. In Section 4.1, we study an image denoising example and in Section 4.2 we consider the Ising spin glass – a model in statistical mechanics that has been used as a benchmark for inference in graphical models.

Our main objective is to demonstrate that Partial SOS can be used successfully on large-scale graphical models, and is competitive with the following popular inference methods:

- **Belief Propagation - Sum Product (BP-SP)**: Pearl's belief propagation computes exact marginal distributions on trees [Pea86]. Given a graph structured objective function $U(\boldsymbol{x})$, we apply BP-SP to the Gibbs-Boltzmann distribution $p(\boldsymbol{x}) = \exp\{U(\boldsymbol{x})\}/Z$ using the standard sum-product update rules with an inertia of $0.5$ to help convergence [YFW05], and threshold the marginals at $0.5$.
- **Belief Propagation - Max Product (BP-MP)**: By replacing the marginal probabilities in the sum-product updates with max-marginals, we obtain BP-MP, which can be used for exact inference on trees [MM09]. For general graphs, BP-MP is closely related to an LP relaxation of the combinatorial problem INT [YFW05, WF01]. Similar to BP-SP, we use an inertia of $0.5$. Note that the Max-Product updates can be equivalently written as Min-Sum updates [MM09].
- **Generalized Belief Propagation (GBP)**: The decision variables in GBP are beliefs (joint probability distributions) over larger subsets of variables in the graph $G$, and they are updated in a message passing fashion [YFW00, YFW05]. We use plaquettes in the grid (contiguous groups of four vertices) as the largest regions, and apply message passing with inertia $0.1$ [WF01].
- **Partial SOS - Degree 2 (PSOS(2))**: By defining regions as single vertices and enforcing only the sphere constraints, we recover the classical Goemans-Williamson SDP relaxation [GW95]. Non-convex Burer-Monteiro approach is extremely efficient in this case [BM03]. We round the SDP solution by $\hat{x}_i = \text{sign}(\langle \boldsymbol{\sigma}_i, \boldsymbol{\sigma}_\emptyset \rangle)$ which is closely related to the classical approach of [GW95].
- **Partial SOS - Degree 4 (PSOS(4))**: This is the algorithm developed in the present paper. We take the regions $R_\ell$ to be triangles, cf. Figure 1, right frame. In an $\sqrt{n} \times \sqrt{n}$ grid, we have $2(\sqrt{n} - 1)^2$ such regions resulting in $\mathcal{O}(n)$ constraints. In Figures 3 and 4, PSOS(4) refers to the CLAP rounding scheme applied together with PSOS(4) in the lift procedure.

### 4.1 Image Denoising via Markov Random Fields

Given a $\sqrt{n} \times \sqrt{n}$ binary image $\boldsymbol{x}_0 \in \{+1, -1\}^n$, we generate a corrupted version of the same image $\boldsymbol{y} \in \{+1, -1\}^n$. We then try to denoise $\boldsymbol{y}$ by maximizing the following objective function:

$$U(\boldsymbol{x}) = \sum_{(i,j) \in E} x_i x_j + \theta_0 \sum_{i \in V} y_i x_i \,, \tag{4.1}$$

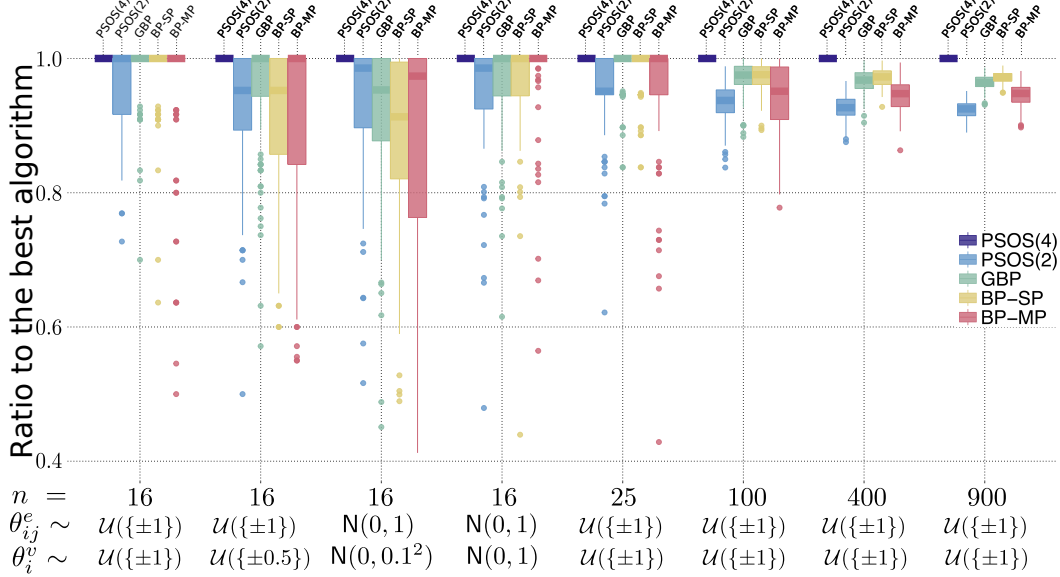

Figure 4: Solving the MAP inference problem INT for Ising spin glasses on two-dimensional grids. $\mathcal{U}$ and N represent uniform and normal distributions. Each bar contains 100 independent realizations. We plot the ratio between the objective value achieved by that algorithm and the exact optimum for $n \in \{16, 25\}$, or the best value achieved by any of the 5 algorithms for $n \in \{100, 400, 900\}$.

where the graph $G$ is the $\sqrt{n} \times \sqrt{n}$ grid, i.e., $V = \{i = (i_1, i_2) : \quad i_1, i_2 \in \{1, \ldots, \sqrt{n}\}\}$ and $E = \{(i,j) : \quad \|i - j\|_1 = 1\}$. In applying Algorithm 1, we add diagonals to the grid (see right plot in Figure 1) in order to satisfy the condition (3.1) with corresponding weight $\theta_{ij}^e = 0$.

In Figure 3, we report the output of various algorithms for a $100 \times 100$ binary image. We are not aware of any earlier implementation of SOS(4) beyond tens of variables, while PSOS(4) is applied here to $n = 10,000$ variables. Running times for CLAP rounding scheme (which requires several runs of PSOS(4)) are of order an hour, and are reported in Figure 3. We consider two noise models: i.i.d. Bernoulli noise and blockwise noise. The model parameter $\theta_0$ is chosen in each case as to approximately optimize the performances under BP denoising. In these (as well as in 4 other experiments of the same type reported in the supplement), PSOS(4) gives consistently the best reconstruction (often tied with GBP), in reasonable time. Also, it consistently achieves the largest value of the objective function among all algorithms.

## 4.2   Ising Spin Glass

The Ising spin glass (also known as Edwards-Anderson model [EA75]) is one of the most studied models in statistical physics. It is given by an objective function of the form INT with $G$ a $d$-dimensional grid, and i.i.d. parameters $\{\theta_{ij}^e\}_{(i,j)\in E}$, $\{\theta_i^v\}_{i\in V}$. Following earlier work [YFW05], we use Ising spin glasses as a testing ground for our algorithm. Denoting the uniform and normal distributions by $\mathcal{U}$ and N respectively, we consider two-dimensional grids (i.e. $d = 2$), and the following parameter distributions: $(i)$ $\theta_{ij}^e \sim \mathcal{U}(\{+1, -1\})$ and $\theta_i^v \sim \mathcal{U}(\{+1, -1\})$, $(ii)$ $\theta_{ij}^e \sim \mathcal{U}(\{+1, -1\})$ and $\theta_i^v \sim \mathcal{U}(\{+1/2, -1/2\})$, $(iii)$ $\theta_{ij}^e \sim$ N$(0, 1)$ and $\theta_i^v \sim$ N$(0, \sigma^2)$ with $\sigma = 0.1$ (this is the setting considered in [YFW05]), and $(iv)$ $\theta_{ij}^e \sim$ N$(0, 1)$ and $\theta_i^v \sim$ N$(0, \sigma^2)$ with $\sigma = 1$. For each of these settings, we considered grids of size $n \in \{16, 25, 100, 400, 900\}$.

In Figure 4, we report the results of 8 experiments as a box plot. We ran the five inference algorithms described above on 100 realizations; a total of 800 experiments are reported in Figure 4. For each of the realizations, we record the ratio of the achieved value of an algorithm to the exact maximum (for $n \in \{16, 25\}$), or to the best value achieved among these algorithms (for $n \in \{100, 400, 900\}$). This is because for lattices of size 16 and 25, we are able to run an exhaustive search to determine the true maximizer of the integer program. Further details are reported in the supplement.

*In every single instance of 800 experiments,* PSOS(4) *achieved the largest objective value, and whenever this could be verified by exhaustive search (i.e. for $n \in \{16, 25\}$) it achieved an exact maximizer of the integer program.*

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
