[Supplementary Material]

# A  Proof of Theorem 1

Given the graph $G = (V, E)$, and parameters $\theta^{\mathrm{e}}, \theta^{\mathrm{v}}$, we can construct a new graph by adding the extra vertex $\emptyset$, together with edges $\{(i, \emptyset) : i \in V\}$ connecting it to all previous vertices, and edge parameters $\theta^{\mathrm{e}}_{i,\emptyset} = \theta^{\mathrm{y}}_i$ (while setting to 0 the vertex parameters). Therefore, one can always eliminate the linear term and work with the quadratic form.

We define the *cut polytope* as

$$\mathcal{C} := \mathrm{Conv}\left(\left\{ xx^T : x_i^2 = 1 \ \forall i \in V \right\}\right), \tag{A.1}$$

which is a convex hull of $2^n$ rank-1 matrices. Introducing the interaction variables $X_{ij} = x_i x_j$, the original optimization problem can be written without the linear term as

$$\underset{X \in \mathbb{R}^{n \times n}}{\text{maximize}} \sum_{(i,j) \in E} W_{ij} X_{ij} \tag{A.2}$$

$$\text{subject to: } X \in \mathcal{C}.$$

For an edge $e = (i, j)$, denote by $X_e$ the entry $X_{ij}$, and for an edge set $F \subset E$ let $X(F)$ be the summation of entries $X_{ij}$ for which $(i, j) \in F$, i.e. $X(F) = \sum_{e \in F} X_e$. Further, define the *metric polytope* as

$$\mathcal{M} := \{ M \in \mathbb{S}^n : |M_e| \leq 1 \ \forall e \in E, \tag{A.3}$$

$$M(F) - M(C \setminus F) \geq 2 - |C| \text{ for } F \subset C, |F| \text{ is odd}, C \text{ is a simple cycle} \}.$$

The inequalities that define the metric polytope are called *cyclic inequalities*. We recall the following result of Barahona and Mahjoub.

**Theorem 2** (Barahona and Mahjoub [BM86]). $G = (V, E)$ *is not contractible to* $\mathcal{K}_5$ *if and only if the cut polytope* $\mathcal{C}$ *is defined by the metric polytope* $\mathcal{M}$.

The above result implies that the cut polytope is defined by the metric polytope, if the underlying graph is not contractible to $K_5$. However, cyclic inequalities are not sufficient to describe $K_5$.

*Proof of Theorem 1.* Define the symmetric matrix $M \in \mathbb{R}^{n \times n}$ as $M_{ij} = M_{ji} = \langle \boldsymbol{\sigma}_i, \boldsymbol{\sigma}_j \rangle$ and $M_{ii} = 1$ for $i, j \in [n]$. Clearly, $M$ is positive semidefinite. Since $\mathcal{R} = \{R_1, R_2, ..., R_m\}$ is a covering of $G$, for each vertex $i \in V$, there exists $j \in [m]$ such that $i \in R_j$. We have $\Sigma(R_j)$ satisfying degree-4 SOS constraints, which implies that relaxed variable $\boldsymbol{\sigma}_i$ is on the unit sphere. Therefore, the entries of $M$ satisfy

$$|M_{ij}| = |\langle \boldsymbol{\sigma}_i, \boldsymbol{\sigma}_j \rangle| \leq \|\boldsymbol{\sigma}_i\| \|\boldsymbol{\sigma}_j\|, \tag{A.4}$$

$$\leq 1,$$

by the Cauchy-Schwartz inequality. Similarly for an edge $(i, j)$, there exists $k \in [m]$ such that $i$ and $j$ both belong to $R_k$. Therefore, the variable $\boldsymbol{\sigma}_{ij}$ satisfies degree-4 SOS constraints, which in turn implies that it is on the unit sphere.

Let $C = \{e_1, e_2, ..., e_N\}$ be a chordless cycle of length $N$ such that $e_1$ and $e_N$ share a common vertex. There exists $p \in [m]$ such that each node defining the elements of $C$ belongs to the region $R_p$. Assume that the nodes $i, j, k \in R_p$. Then,

$$M_{ij} = \langle \boldsymbol{\sigma}_i, \boldsymbol{\sigma}_j \rangle = \langle \boldsymbol{\sigma}_{ij}, \boldsymbol{\sigma}_0 \rangle, \tag{A.5}$$

by the undirected constraints. Moreover, by using the triangle constraints we can write

$$0 \leq \frac{1}{4} \|\boldsymbol{\sigma}_{ij} + \boldsymbol{\sigma}_{jk} - \boldsymbol{\sigma}_{ik} - \boldsymbol{\sigma}_0\|^2, \tag{A.6}$$

$$= 1 + \frac{1}{2}\langle \boldsymbol{\sigma}_{ij}, \boldsymbol{\sigma}_{jk} \rangle - \frac{1}{2}\langle \boldsymbol{\sigma}_{ij}, \boldsymbol{\sigma}_{ik} \rangle - \frac{1}{2}\langle \boldsymbol{\sigma}_{ij}, \boldsymbol{\sigma}_0 \rangle - \frac{1}{2}\langle \boldsymbol{\sigma}_{jk}, \boldsymbol{\sigma}_{ik} \rangle - \frac{1}{2}\langle \boldsymbol{\sigma}_{jk}, \boldsymbol{\sigma}_0 \rangle + \frac{1}{2}\langle \boldsymbol{\sigma}_{ik}, \boldsymbol{\sigma}_0 \rangle,$$

$$= 1 + \langle \boldsymbol{\sigma}_{ik}, \boldsymbol{\sigma}_0 \rangle - \langle \boldsymbol{\sigma}_{ij}, \boldsymbol{\sigma}_0 \rangle - \langle \boldsymbol{\sigma}_{jk}, \boldsymbol{\sigma}_0 \rangle$$

and similarly,

$$0 \leq \frac{1}{4} \|\boldsymbol{\sigma}_{ij} + \boldsymbol{\sigma}_{jk} + \boldsymbol{\sigma}_{ik} + \boldsymbol{\sigma}_0\|^2, \tag{A.7}$$

$$= 1 + \langle \boldsymbol{\sigma}_{ik}, \boldsymbol{\sigma}_0 \rangle + \langle \boldsymbol{\sigma}_{ij}, \boldsymbol{\sigma}_0 \rangle + \langle \boldsymbol{\sigma}_{jk}, \boldsymbol{\sigma}_0 \rangle.$$

Using these two inequalities, we can conclude that $\forall i, j, k \in R_p$,

$$|\langle \boldsymbol{\sigma}_{ij}, \boldsymbol{\sigma}_0 \rangle + \langle \boldsymbol{\sigma}_{jk}, \boldsymbol{\sigma}_0 \rangle| \leq 1 + \langle \boldsymbol{\sigma}_{ik}, \boldsymbol{\sigma}_0 \rangle,$$
$$\implies |M_{ij} + M_{jk}| \leq 1 + M_{ik}. \tag{A.8}$$

Next, we will show that $M$ satisfies the cyclic inequalities given in Eq. (A.3). Recall that $C$ is a chordless cycle $C = \{e_1, e_2, ..., e_N\}$ of $G$, and let edges forming $C$ be given as $e_i = (v_i, v_{i+1})$ for $i \in [N]$, and $v_{N+1} = v_1$. Let $F \subset C$ be a set of edges with odd cardinality. There is at least one edge belonging $F$. We will denote by $e_{i\triangle}$, the edge created by joining $v_1$ and $v_i$. Note that $e_{2\triangle} = e_1$ and $e_{N\triangle} = e_N$. For the simple cycle $C$, by adding the edges $\{e_{3\triangle}, e_{4\triangle}, ..., e_{N-1\triangle}\}$ we have created $N - 3$ chords to construct the chordal graph of $C$, where $e_i$, $e_{i\triangle}$ and $e_{i+1\triangle}$ form a triangle.

Let $s_j \in \{-1, +1\}$ be the indicator variable for $e_j$'s membership to the set $F$ ($s_j = 1$ if $e_j \in F$). We have $\prod_{i=1}^{N} s_i = (-1)^{N-|F|}$ which implies that $s_N = (-1)^{N-|F|} \prod_{i=1}^{N-1} s_i$. Finally, we let $s_{i\triangle} = \prod_{j=1}^{i-1} s_j$ for $i \geq 2$ and observe that $s_{i+1\triangle} = s_{i\triangle} s_{i+1}$. Noticing that

$$M(F) - M(C \setminus F) = \sum_{i=1}^{N} s_i M_{e_i}, \tag{A.9}$$

we write the following inequalities that are based on the triangle inequalities given in Eq. (A.8),

$$s_1 M_{e_1} + s_2 M_{e_2} + s_{3\triangle} M_{e_{3\triangle}} + 1 \geq 0, \tag{A.10}$$
$$s_3 M_{e_3} - s_{3\triangle} M_{e_{3\triangle}} - s_{4\triangle} M_{e_{4\triangle}} + 1 \geq 0,$$
$$s_4 M_{e_4} + s_{4\triangle} M_{e_{4\triangle}} + s_{5\triangle} M_{e_{5\triangle}} + 1 \geq 0,$$

$$\vdots \qquad\qquad \vdots$$

$$s_{N-1} M_{e_{N-1}} + (-1)^{N-1} s_{N-1\triangle} M_{e_{N-1\triangle}} + (-1)^{N-1} s_{N\triangle} M_{e_{N\triangle}} + 1 \geq 0$$

By summing these inequalities, we obtain that

$$\sum_{i=1}^{N-1} s_i M_{e_i} + (-1)^{N-1} s_{N\triangle} M_{e_{N\triangle}} + N - 2 \geq 0. \tag{A.11}$$

Since we also have $s_{N\triangle} = \prod_{i=1}^{N-1} s_i = s_N (-1)^{N-|F|}$ we can write

$$(-1)^{N-1} s_{N\triangle} = s_N (-1)^{2N-|F|-1} = s_N \tag{A.12}$$

since $|F|$ is odd. Therefore the inequality in Eq. (A.11) reduces to

$$\sum_{e \in F} M_e - \sum_{e \in C \setminus F} M_e \geq 2 - N. \tag{A.13}$$

This implies that $M \in \mathcal{M}$. Finally, we invoke the result given in Theorem 2 and conclude the proof.

$\square$

# B  Additional Experiments

Figure 5: Additional denoising experiments of a binary image by maximizing the objective function Eq. (4.1). First 4 rows: i.i.d. Bernoulli error with flip probability $p \in \{0.05, 0.1, 0.15, 0.2\}$ with $\theta_0 = 1.26$. Last 2 rows: blockwise noise where each pixel is the center of a $3 \times 3$ error block independently with probability $p \in \{0.006, 0.01\}$ and $\theta_0 = 1$. Final objective value attained by each algorithm along with its run time is reported under each image. We observe that PSOS(4) achieves the best objective value compared to the other inference algorithms.

## C   Further Details of the Experiments in Section 4.2

Table 1: Details of the experiments shown in Figure 4. We report statistics of run-time and ratio to the best algorithm. More specifically, we report the mean and standard deviation of the run-time of each algorithm within each experiment (100 replications). We also report $\%5/\%10/\%60$ quantiles of the ratio of the objective value achieved by an algorithm and the exact optimum for $n \in \{16, 25\}$, or the best value achieved by any of the 5 algorithms for $n \in \{100, 400, 900\}$.

| EXPERIMENT↓ | STATS↓ | PSOS-4 | PSOS-2 | GBP | BP-MP | BP-SP |
|---|---|---|---|---|---|---|
| $n = 16$ $\theta_i^v \sim \mathcal{U}(\pm 1)$ $\theta_{ij}^e \sim \mathcal{U}(\pm 1)$ | TIME(MEAN/SD): RATIO(5/10/60% QT) | 3.9/.3 1./1./1. | .2/.0 .82/.83/1. | 2.8/1.6 .91/.92/1. | 1./.8 .71/.82/1. | 1./.4 .91/.91/1. |
| $n = 16$ $\theta_i^v \sim \mathcal{U}(\pm .5)$ $\theta_{ij}^e \sim \mathcal{U}(\pm 1)$ | TIME(MEAN/SD): RATIO(5/10/60% QT) | 4./.3 1./1./1. | .2/.0 .71/.79/1. | 3.7/1.7 .78/.83/1. | 1.8/.9 .57/.65/1. | 1.7/.5 .65/.74/1. |
| $n = 16$ $\theta_i^v \sim \mathsf{N}(0, .01)$ $\theta_{ij}^e \sim \mathsf{N}(0, 1)$ | TIME(MEAN/SD): RATIO(5/10/60% QT) | 4.4/.5 1./1./1. | .2/.0 .72/.83/1. | 3.8/2. .67/.76/.97 | 2.1/.7 .41/.5/.98 | 1.4/.4 .52/.69/.95 |
| $n = 16$ $\theta_i^v \sim \mathsf{N}(0, 1)$ $\theta_{ij}^e \sim \mathsf{N}(0, 1)$ | TIME(MEAN/SD): RATIO(5/10/60% QT) | 4./.3 1./1./1. | .2/.0 .78/.87/1. | 2.5/1.4 .8/.86/1. | .9/.33 .83/.96/1. | .81/.3 .83/.89/1. |
| $n = 25$ $\theta_i^v \sim \mathcal{U}(\pm 1)$ $\theta_{ij}^e \sim \mathcal{U}(\pm 1)$ | TIME(MEAN/SD): RATIO(5/10/60% QT) | 7.1/.8 1./1./1. | .3/.0 .84/.87/1. | 9./3.2 .9/.95/1. | 2.4/1.6 .73/.84/1. | 1.9/.8 .9/.95/1. |
| $n = 100$ $\theta_i^v \sim \mathcal{U}(\pm 1)$ $\theta_{ij}^e \sim \mathcal{U}(\pm 1)$ | TIME(MEAN/SD): RATIO(5/10/60% QT) | 58./10. 1./1./1. | 1.3/.1 .87/.89/.94 | 77.7/.4 .92/.94/.99 | 17.7/3.9 .85/.87/.96 | 14.4/2.4 .93/.94/.99 |
| $n = 400$ $\theta_i^v \sim \mathcal{U}(\pm 1)$ $\theta_{ij}^e \sim \mathcal{U}(\pm 1)$ | TIME(MEAN/SD): RATIO(5/10/60% QT) | 360.3/83.4 1./1./1. | 5.7/.4 .89/.9/.93 | 386.8/7. .93/.94/.97 | 83.3/.7 .9/.91/.95 | 69.4/.5 .95/.96/.98 |
| $n = 900$ $\theta_i^v \sim \mathcal{U}(\pm 1)$ $\theta_{ij}^e \sim \mathcal{U}(\pm 1)$ | TIME(MEAN/SD): RATIO(5/10/60% QT) | 757.4/108. 1./1./1. | 13.9/1.1 .9/.91/.93 | 939.8/31.4 .94/.95/.97 | 194./1.4 .91/.92/.95 | 161.8/1.3 .95/.96/.97 |