[Reviews · NeurIPS 2017]

Reviewer 1



The authors show how SDP methods based on the Sum-of-Squares (SOS) hierarchy may be used for MAP inference in discrete graphical models, focusing on binary pairwise models. Specifically, an approximate method for binary pairwise models is introduced to solve what is called PSOS(4), then the solution is rounded to an integer solution using a recursive scheme called CLAP (for Confidence Lift And Project). Preliminary empirical results are presented which appear encouraging. This is an interesting direction but I was confused by several aspects. Please could the authors clarify/note the following: 1. l.60-67 Optimizing an LP over a relaxation in the Sherali-Adams hierarchy, and relation to Theorem 1. For the first order relaxation, i.e. the standard pairwise local polytope, only edges in the original model need be considered. This is because, in this case, it is straightforward to observe that consistent marginals for the other edges (which do not exist in the original model) exist while remaining in the pairwise polytope for the complete graph. By asusmption their values do not affect the optimum. For the next higher order relaxation, i.e. the triplet-consistent polytope TRI, in fact again it may be shown *for binary pairwise models whose graph is chordal* that one need only consider triangles in the original model, since there always exist consistent marginals for all other edges. This is not obvious and was shown recently by Rowland, Pacchiano and Weller "Conditions beyond treewidth..." AISTATS 2017, Sec 3.3. This theme is clearly relevant to your approach, as described in l.81-89. Further, properties of TRI (equivalent for binary pairwise models to the cycle polytope, see Sontag PhD thesis 2010) may be highly relevant for this paper since I believe that the proof for Theorem 1 relies on showing essentially that the SOS(4) relaxation lies within TRI - please could you comment? If that is true, is it true for any binary pairwise model or only subject to certain restrictions? 2. l.72 Here you helpfully suggest that SDP relaxations do not account well for correlations between neighboring vertices. It would be very valuable if you could please elaborate on this, and other possible negatives of your approach, to provide a balanced picture. Given this observation, why then does your method appear to work well in the empirical examples you show later? 3. I appreciate space constraints are very tight but if possible to elaborate (even in the Appendix) on the approach, it would be very helpful. In particular: what exactly is happening when r is varied for a given model; how should we think about the directed i->j constraints in l.124-5? 4. Alg 1. When will this algorithm perform well/badly? Can any guarantees be provided, e.g. will it give an upper bound on the solution? 5. CLAP Alg 2. Quick check: is this guaranteed to terminate? [maybe for binary pairwise models, is < \sigma_0, \sigma_s > always \geq 1/2 or some other bound?] Could you briefly discuss its strengths and weaknesses. Are other methods such as Barak, Kelner, Steuer 2014 "Rounding sum-of-squares relaxations" relevant? 6. Sec 4 Experiments. When you run BP-SP, you obtain marginals. How do you then compute your approximate MAP solution? Do you use the same CLAP rounding approach or something else? This may be important since in your experiments, BP-SP performs very well. Since you use triangles as regions for PSOS(4), could you try the same for GBP to make the comparison more similar? Particularly since it appears somewhat odd that the current GBP with 4-sets is not doing better than BP-SP. Times should be reported for all methods to allow more meaningful comparisons [I recognize this can be tricky with non-optimized code but the pattern as larger models are examined would still be helpful]. If possible, it would be instructive to add experiments for larger planar models with no singleton potentials, where it is feasible to compute the exact MAP score. Minor points: In a few places, claims are perhaps stronger than justified - e.g. in the Abstract, "significantly outperforms BP and GBP" ; l. 101 perhaps remove "extensive"; l. 243 - surely the exact max was obtained only for the small experiments; you don't know for the larger models? A few capitalizations are missing in the References, e.g. Ising, Burer-Monteiro, SDP ======================= I have read the rebuttal and thank the authors for addressing some of my concerns.

Reviewer 2



Summary: The authors observe that SDP is not used for MAP inference due to its inability to model expressive local interactions between variables and due to the poor scalability of the problem formulation. They address the first limitation by introducing a partial sum-of-squares relaxation which allows regional interactions between variables, with regions formulated as a vertex cover. To address the second limitation, the authors introduce an approximate optimizer using an augmented Lagrangian for constraints, and then project the approximate relaxed solution into a feasible space. Experiments show acceptable scalability and competitive performance. Pros: - Clearly written and organized paper - Increases the applicability of SDP to a broader set of problems and domains Cons: - Seems half-finished, with a minimal theoretical analysis at the end that ends abruptly - Does not discuss scalability-expressivity tradeoffs in PSOS assumptions, nor the approximation quality of the optimization procedure Detailed Comments: This paper is clearly written and makes a case for a set of limitations of current work and a set of innovations that address these limitations. I have difficulty gauging the depth of these contributions, and the paper would benefit from a more nuanced discussion of the tradeoffs involved (beyond performance plateauing after r=10, for example). I was surprised by the abrupt ending of the paper, and felt that a longer treatment of the theoretical analysis would be more coherent, particularly given the puzzling performance of PSOS.

Reviewer 3



In general I like the paper and recommend acceptance. However this is not a clear cut win. There are a number of caveats in the paper. Fixing these may help the readers to appreciate contribution of the paper better. Here are my comments on what would be good to correct/improve a) Only binary model is discussed in the paper and one discovers it only on ln.108. b) The resulting algorithm is comapred with sum-product version of BP (message-passing), in spite of the fact that the suggested algorithm solves the optimizatiom (maximum likelihood) optimization (not inference) problem. It would be more appropriate to compare the algorithm with min-sum message-passing algorithm.